# Coupling Dilated Encoder–Decoder Network for Multi-Channel Airborne LiDAR Bathymetry Full-Waveform Denoising

Bin Hu [1,2], Yiqiang Zhao [1,*], Guoqing Zhou [3], Jiaji He [1], Changlong Liu [4], Qiang Liu [1], Mao Ye [1] and Yao Li [1]

1    School of Microelectronics, Tianjin University, Tianjin 300072, China; hubin@email.tjut.edu.cn (B.H.);
     dochejj@tju.edu.cn (J.H.); qiangliu@tju.edu.cn (Q.L.); mao_ye@tju.edu.cn (M.Y.); liyao@tju.edu.cn (Y.L.)
2    Technical College for the Deaf, Tianjin University of Technology, Tianjin 300382, China
3    College of Geomatics and Geoinformation, Guilin University of Technology, Guilin 541004, China;
     gzhou@glut.edu.cn
4    Fifty-Fourth Research Institute of China Electronics Technology Group Corporation,
     Shijiazhuang 050081, China; liuclcti@163.com
*    Correspondence: yq_zhao@tju.edu.cn

**Abstract:** Multi-channel airborne full-waveform LiDAR is widely used for high-precision underwater depth measurement. However, the signal quality of full-waveform data is unstable due to the influence of background light, dark current noise, and the complex transmission process. Therefore, we propose a nonlocal encoder block (NLEB) based on spatial dilated convolution to optimize the feature extraction of adjacent frames. On this basis, a coupled denoising encoder–decoder network is proposed that takes advantage of the echo correlation in deep-water and shallow-water channels. Firstly, full waveforms from different channels are stacked together to form a two-dimensional tensor and input into the proposed network. Then, NLEB is used to extract local and nonlocal features from the 2D tensor. After fusing the features of the two channels, the reconstructed denoised data can be obtained by upsampling with a fully connected layer and deconvolution layer. Based on the measured data set, we constructed a noise–noisier data set, on which several denoising algorithms were compared. The results show that the proposed method improves the stability of denoising by using the inter-channel and multi-frame data correlation.

**Keywords:** full-waveform denoising; deep learning; airborne LiDAR; bathymetry

## 1. Introduction

Airborne full-waveform LiDAR is a type of laser ranging system that can store and record the complete reflected echo signal of the target in a very short sampling interval and with a large capacity, so it can obtain a high-range resolution [1]. The scanning modes of airborne LiDAR detection systems mainly include linear scanning and oval scanning. The advantage of linear scanning is that the system can maximize the acquisition of the light energy of the normal incident and nearby angles of the target, but the disadvantage is that the structure is complex and requires high synchronization accuracy. The Airborne LiDAR Bathymetry (ALB) system described in this paper adopts an ovoid scanning mode with a light structure; the incidence angle of the scanning mode is relatively stable, which is conducive to avoiding the direct reflection of the surface mirror and reducing differences in echo energy [2,3].

Airborne LiDAR is widely used in land cover classification [4], marine resource detection [5,6], and vegetation detection [7] because of its high measuring efficiency, wide working range, high measuring accuracy, and ability to obtain the physical characteristics of the target. Because 532 nm blue-green light has good water permeability, laser radar at this wavelength is used to measure the depth of the seafloor. Since the laser beam from the atmosphere enters through the sea surface and travels through the water, the energy returned to the detector will be reduced, which will reduce the peak signal-to-noise ratio of

the seafloor signal and increase the uncertainty of the distance measurement [8]. The full waveform collected by the ALB system can be regarded as composed of the sea-surface, water scattering, and seabed signals. Sea-surface signals are easily lost due to the influence of waves, the scattered part of the water scattering signal is relatively stable, and weak seafloor echoes are affected by the terrain, water quality, and other factors, so the integrity of the signal needs to be repaired through multi-frame correlation.

The maximum measured depth and accuracy are also affected by the water quality, target reflectance, background noise, and flight altitude [9]. As shown in Figure 1, the multi-field-of-view detection structure is widely used in airborne full-waveform sounding systems, and the multi-channel data in the field of view can be used to improve the dynamic range and accuracy of detection [5]. The small field in the middle is used as a shallow-water channel to obtain the energy-concentrated shallow-water echo, while the remaining external field of view is used as a deep channel to receive weak submarine echo signals in deep water owing to a stronger gain and a larger field of view.

Different field-of-view echoes have similar structures, so the target characteristics reflected by multi-channel signals can be used to optimize the target peak signal-to-noise ratio.

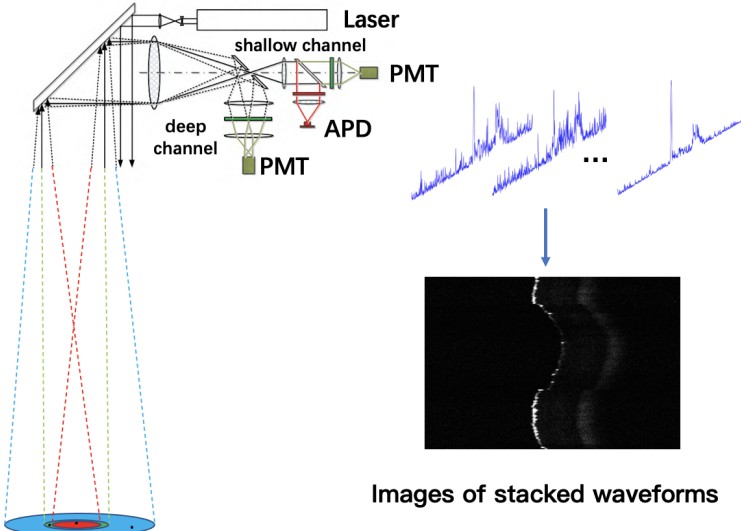

**Figure 1.** Introduction of multi-channel full-waveform airborne LiDAR.

Although the hardware architecture of airborne LiDAR has been improved, the process of the laser light returning to the detector after traveling through the atmosphere, the air–sea interface, and the water is extremely complex.

Random background light can produce Gaussian noise, and sudden changes in the seafloor topography or the scattering of water can produce sparse noise. Therefore, the mixed noise contained in the full waveform of ALB has a great influence on the seabed signal. Traditional algorithms, such as wavelet filtering [10], EMD filtering [11], waveform decomposition [12], and deconvolution [13], are widely used in this field. However, the effect of such algorithms is often limited by manually adjusted parameters. Full-waveform data can be regarded as a one-dimensional time series, so deep learning is widely used in LiDAR data processing [14,15]. Super-resolution technology can restore the echo with a higher sampling rate without upgrading the hardware [16,17]. Although multi-channel information has been proven to be effective for echo classification, denoising and signal enhancement have not been used [5,6]. Due to the structural similarity between consecutive waveforms and the physical correlation between different channels of the same frame [18], they can be used to further improve the signal sampling rate and reduce noise [19]. However, with the increase in the detection distance, the intensity of the LiDAR echo signal steadily decreases. To solve the above problems, this paper will further explore

the influence of the signal correlation between channels on full-waveform denoising. The main contributions of this paper are as follows:

- Based on a multi-task deep learning architecture and a centroid alignment algorithm, we propose a novel LiDAR denoising technique for multi-channel ALB systems that can improve the stability of denoising.
- The NLEB module is proposed, and the loss function is optimized to explore the impact of intra-channel autocorrelation and inter-channel structural similarity on LiDAR signal enhancement.
- We analyzed the characteristics of multi-channel ALB data and the limitations of the denoising algorithm by using measured data.

## 2. Related Work

In order to improve the accuracy of the ALB system, traditional algorithms and methods based on deep learning are proposed for filtering, echo classification, and resolution enhancement.

### 2.1. Model-Based Methods

The detection accuracy depends on the position error when the full waveform is decomposed into sub-echo components, and the transmitted waveform is generally considered to be Gaussian-like [20]. Due to the influence of cross-medium transmission, the echo may be affected by trailing, ringing, and other phenomena, resulting in a reduction in ranging accuracy in the subsequent waveform decomposition [21]. Researchers often use the trigonometric function, quadrilateral function, and exponential function to decompose the echo into water surface, water backscatter, and underwater signals [22,23]. For a seabed signal with certain distortion, a model with a generalized Gaussian distribution or lognormal distribution can be used to fit the signal. The multi-Gaussian decomposition method based on initial parameters is not suitable for the peak detection of distorted pulses in extremely shallow water with high overlap [24]. The combination of dynamic threshold and wavelet filtering [8,10] can improve the ability to recognize weak signals, but the selection of the wavelet basis function and decomposition scale need to be adjusted for different water depths. The Richardson–Lucy deconvolution [13,25] algorithm can be used to extract the interface response function to separate multiple targets with a short distance, but the algorithm relies on multiple iterations, which tend to produce noise and sound effects. Therefore, superimposing structurally similar subsequences in multiple frames [19] can significantly improve the peak signal-to-noise ratio of submarine signals. B-spline [26] and Wiener deconvolution algorithms [27] require no prior knowledge and avoid short-distance stacking losses, but they are too sensitive to noise. Neighborhood and multidimensional space–time information [28] can be used to improve the peak signal-to-noise ratio of seafloor signals, and waveform superposition can improve the detection rate of weak seafloor signals. However, parameter debugging in traditional algorithms still limits the processing performance of ALB echoes [29]. The existing waveform processing methods do not make full use of the correlation between multi-frame data, and stacking the average results of adjacent frames lacks stability in the ALB system [30].

### 2.2. Learning-Based Methods

With the development of deep learning in the field of time-series data processing, researchers have become inclined to use deep learning to automatically extract full-waveform features [15,17] from data. Unsupervised denoising based on an encoder–decoder has achieved good results [14,31]. The deconvolution layer is used for upsampling, and high-quality echo signals can be obtained at a low sampling rate [16,32]. By combining the prior knowledge provided by echo classification with the denoising algorithm [33], the parameter selection strategy can be optimized for different scenes, which can optimize the peak signal-to-noise ratio of the target. Artificial features such as the echo width, skewness, and kurtosis, combined with machine learning methods such as support vector

machines [7] and random forest models [34], can be used to distinguish between deep-water, shallow-water, and land echoes. However, hand-extracted features require specialized knowledge and extensive preprocessing. The feature weights of key positions can be optimized by converting the airborne waveform into a time-series diagram and extracting the high-dimensional temporal and spatial features of the Gramian angular field time-series diagram using the mature network architecture in the image field [35,36]. Based on the LSTM module and one-dimensional convolution module, the time sequence dependence and local spatial features of signals can be extracted [37,38]. Since sea-surface and seabed signals are not fixed in each frame signal, the denoising effect of ALB signals is limited. The seabed topography is complicated and easily affected by impulse noise, so it is necessary to introduce a nonlocal [39] processing module or use multi-frame features to alleviate the problem of the model falling into the local optimal solution.

## 3. Materials

### 3.1. Analysis of ALB Data

ALB data mainly include the sea-surface echo, water backscatter, and seabed echo. With the increase in the laser transmission distance in the sea, the backscatter will be broadened, which will affect the shape and position of the sea-bottom echo. Multi-channel airborne LiDAR uses a high-sensitivity photomultiplier tube to improve the detection of weak sea-bottom signals, but this also leads to greater echo noise. The received power of a sea-surface echo can generally be expressed as follows:

$$P_s = \frac{P_0 \cdot \rho \cdot exp[-2\sigma H] \cdot A \cdot E}{2\pi H^2} \tag{1}$$

where $P_s$ is the sea-surface-reflected power, $P_0$ is laser emission peak power, $\rho$ is the air attenuation coefficient, $A$ is the receiving optical aperture, $E$ is the receiving system efficiency, $H$ is the aircraft flight altitude, and $\sigma$ is the sea-surface reflectivity. The received power of the seabed can generally be expressed as follows:

$$P_b = \frac{P_0 \cdot R \cdot (1-\rho)^2 \cdot exp[-2(\sigma H + \tau D)] \cdot A \cdot E}{2\pi(H + D/n)^2} \tag{2}$$

where $P_b$ is the seabed-reflected power, $R$ is the bottom reflectivity, $\tau$ is the effective attenuation coefficient, and $n$ is the refractive index of seawater.

In order to further analyze the correlation between multi-channel echoes, we used Monte Carlo models to simulate deep- and shallow-water echoes in different states. We monitored a million photons at once that move randomly through the water and disperse using the Henyey–Greenstein function. By varying the scattering attenuation coefficient of seawater to simulate various water conditions, we changed the longest distance of photon propagation as well as the forward distance of photons at each step.

As shown in Figure 2, the x- and y-axes are the time and normalized amplitude, respectively. The red curve is the waveform of the shallow-water channel, and the blue curve is the waveform of the deep-water channel. The main differences between the deep- and shallow-water channels are the field angle and gain. As shown in Figure 2a,b, when the water depth is relatively shallow, obvious sea-surface echoes can be observed in both channels, but shallow-water channels are more sensitive to underwater signals. As shown in Figure 2c,d, the seabed signals of deep channels become more obvious with the increase in water depth, and the data between channels also show a stronger correlation. The echos of deep and shallow channels can be decomposed into the surface, seabed, and backscatter, and the seabed signal gradually weakens and slides to the right until it disappears.

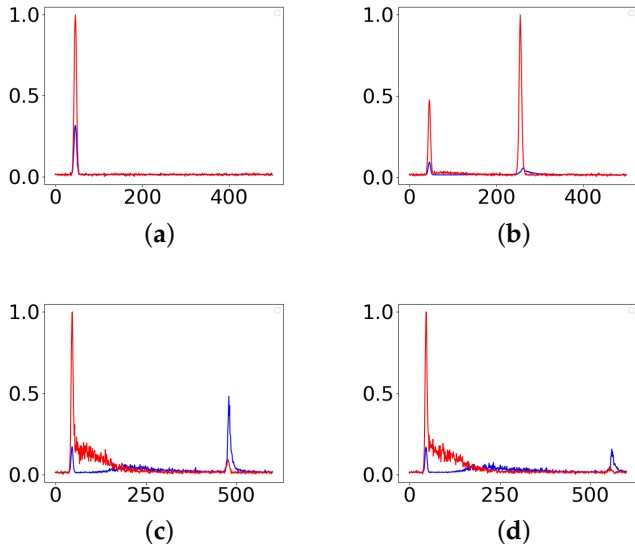

**Figure 2.** Visual comparison of simulation data in deep-water channel and shallow-water channel denoising. (**a**) Extreme shallow-water waveform; (**b**) shallow-water waveform with a strong bottom return; (**c**) deep-water waveform; (**d**) shallow-water waveform with a weak bottom return.

### 3.2. Measured Data Set

As shown in Figure 3, the data in this flight experiment were collected near the west side of Weizhou Island at a flight altitude of 500 m. The echoes of a LiDAR sea survey performed in the evening with good weather were used as the original data set. The flight area was bounded by a longitude of 109°00′ to 109°05′ east and a latitude of 21°01′ to 21°03′ north. The total area of the test area was about 41 km², and the flight width corresponding to the route was 350 m. At the junction of the sea area and the land area, the aircraft flew along 12 north–south navigation belts, each of which was 1.5 km long, and the side overlap rate of the navigation belt was higher than 0.3. In the figure, the navigation belt in red (east–west) was used for system parameter checking and calibration, and the belt in green was used for accuracy checking. To avoid inertial measurement unit (IMU) error accumulation, a figure-of-eight flight was performed before entering the test area. The LiDAR device was mounted on a single-engine turboprop lightweight general-purpose aircraft and is equipped with a positioning and orientation system. The pitch angle and roll angle of the route were generally not more than 2°. The slope of the aircraft did not exceed 22° when turning.

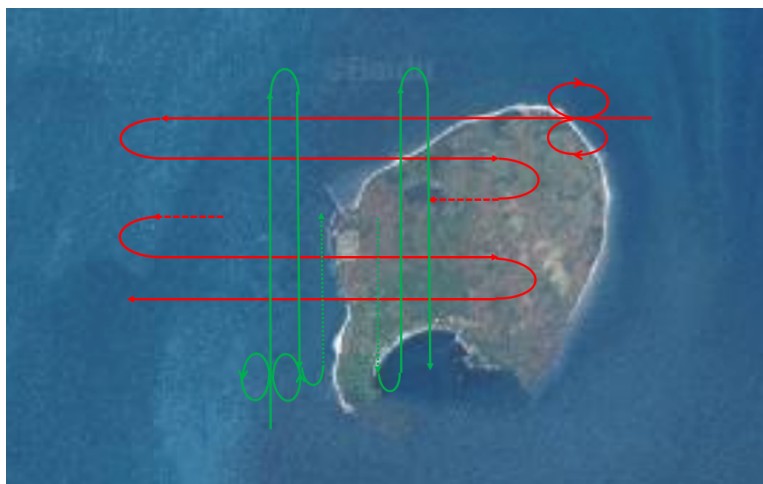

**Figure 3.** Measured scene.

The measured signal cannot provide a pure echo for testing, so we extracted the first 200 points and the last 200 points of the echo to estimate the noise and added them to the original echoes according to different scales to obtain noise and noisier pairs of echoes. We selected continuous echoes to generate a 2D tensor and ensured that the maximum depth change of the echo in each 2D tensor did not exceed 2 m. Considering the speed and angle of view of the aircraft, we stacked 100 frames as a tensor. Therefore, redundant information in multiple frames could be used to eliminate noise. The system parameters of full-waveform LiDAR are shown in Table 1. The water depth in the tested sea area is between 0.3 m and 25 m. The beam divergence angle of the prototype is 0.2 mrad, the scanning angle is 10°, the measurement accuracy of the scanning angle is better than 0.003, the repetition rate is 5.5 kHz, the laser linewidth is 0.05 nm, and the pulse width is 3.3 ns; the aircraft speed was 200 km/h.

**Table 1.** The system parameters of full-waveform LiDAR.

| Parameters | Parameter Indexes |
|---|---|
| Beam divergence angle | 0.2 mrad |
| Laser emission frequency | 5.5 kHz |
| Laser emission power | 17.4 W/532 nm,18 W/1064 nm |
| Pulse energy | 3.05 mJ/532 nm, 3.5 mJ/1064 nm |
| Pulse width | 3.3 ns |
| Detection range | 500–1000 m |
| Scanning mode | Constant-angle conical scan |
| Scanning angle | ±10° |
| Scanning speed | 10 revolutions per second |
| Receiving aperture | 0.2 m |
| Detection minimum energy | $8 \times 10^{-9}$ W |
| Maximum receiving frequency | 2 GHz |

## 4. Proposed Method

In this section, we provide the details of preprocessing, the nonlocal encoder block (NLEB), and the architecture of proposed model.

### 4.1. Preprocessing of Full Waveform

Laser transmission on the sea surface and in the water results in random noise in the full waveform, but the position of the seabed echo is relatively stable in adjacent frames. As shown in Figure 4, by aligning and stacking multiple frames, the similarity of adjacent frames can be highlighted without modifying key information, such as the sea surface, backscatter, and underwater.

To alleviate the impact of abnormal echoes on multi-frame denoising, the echo of the lost sea surface is filtered through the hard threshold of the pulse width and amplitude. Then, the amplitude of the signal is normalized according to the maximum and minimum values of the echoes in each channel. Although the transmitted waveform is a Gaussian pulse with concentrated energy, the sea-surface echo will deform due to the random jitter of the sea surface. The idea of eliminating the echo noise from the ALB system by stacking signals and using related information from multiple echoes has long been proposed to improve the accuracy of seawater depth calculations. Because the water depth does not change much between adjacent frames, echoes in the same field of view have structural similarity [9,40]. Before stacking multi-frame data, we highlight the relative position of the sea surface and seabed through the centroid alignment of the sea surface. The centroid calculation formula is as follows:

$$t_c = \frac{u_1 t_1 + u_2 t_2 + \cdots + u_n t_n}{u_1 + u_2 + \cdots + u_n} = \frac{\sum_{i=1}^{n} u_i t_i}{\sum_{i=1}^{n} u_i} \tag{3}$$

where $t_c$ represents the corrected centroid time, $t_i$ represents the corresponding time of each sampling point of the pulse, and $u_i$ corresponds to the amplitude of each sampling point. Then, we convert 100 waveforms to the same position according to the centroid position of the sea surface to form a two-dimensional tensor.

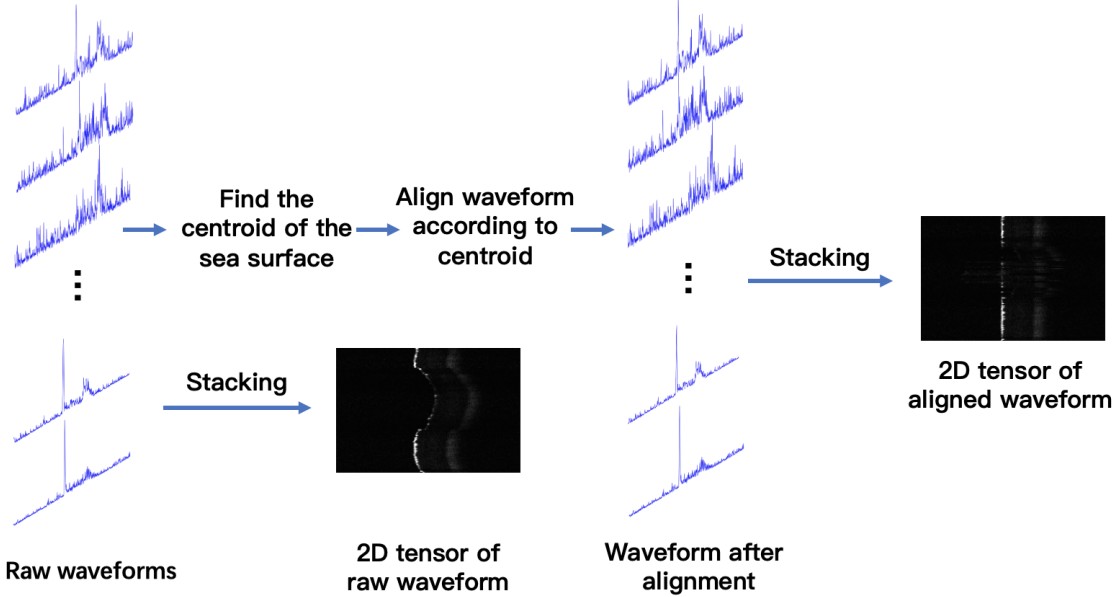

**Figure 4.** The basic principle of full-waveform stacking.

### 4.2. Nonlocal Encoder Block

Due to the impact of background noise, sea-surface jitter, underwater transmission, and other factors on ALB, the received full-waveform signal-to-noise ratio is unstable. However, the sampling rate of the receiving circuit is very high, and the submarine signals covered by the laser in the same field of view are very close, so there is also a correlation between non-adjacent echoes. Using the traditional nonlocal module to extract the distance dependence of non-adjacent echoes will greatly increase the calculation cost [41]. To reduce the consumption of resources and make full use of the spatial correlation between features, we propose an NLEB module with a wide acceptance domain in the spatial domain.

Due to the strong correlation with the close position of the echo in the field of view, context feature extraction can be improved by multi-layer stacked convolution. Including LSTM or using a large-scale convolution kernel will not only enhance the long-distance dependence extraction effect but also add more parameters and decrease computational efficiency. Multiple small-scale dilated convolution stacks are widely used in image denoising, which depends on fewer parameters and increases the nonlinear extraction ability of the model. So, we rely on a convolution kernel size of 3 for stacking, focusing on expanding the spatial-scale sensitivity field. The details of NLEB are shown in Figure 5. It consists of three two-dimensional convolution layers with the kernel size (3,3), and the dilation rate is set to (1,1), (2,1), and (4,1). The kernels are not dilated in the input's time-domain direction. By expanding the convolution of the spatial direction, the receptive field of NLEB can extract features from similar positions between frames without increasing the amount of computation. A batch normalization layer is introduced to accelerate convergence and prevent over-fitting.

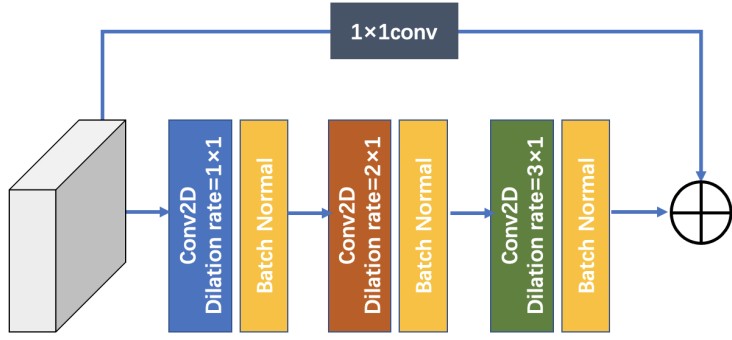

**Figure 5.** The details of nonlocal encoder block.

### 4.3. CNLD-Net Architecture

As previously stated, structural similarity exists between echoes collected in the same area with different gains and fields of view. The idea of eliminating ALB system echo noise by stacking signals and using related information from multiple echoes has long been proposed. To improve the denoising effect, we built a multi-branch network to improve its robustness to weak signals. As shown in Figure 6, our proposed architecture combines NLEB and the idea of multi-task learning. Following the preprocessing process described above, adjacent frames are roughly aligned and stacked according to the sea-surface position before being input into the network.

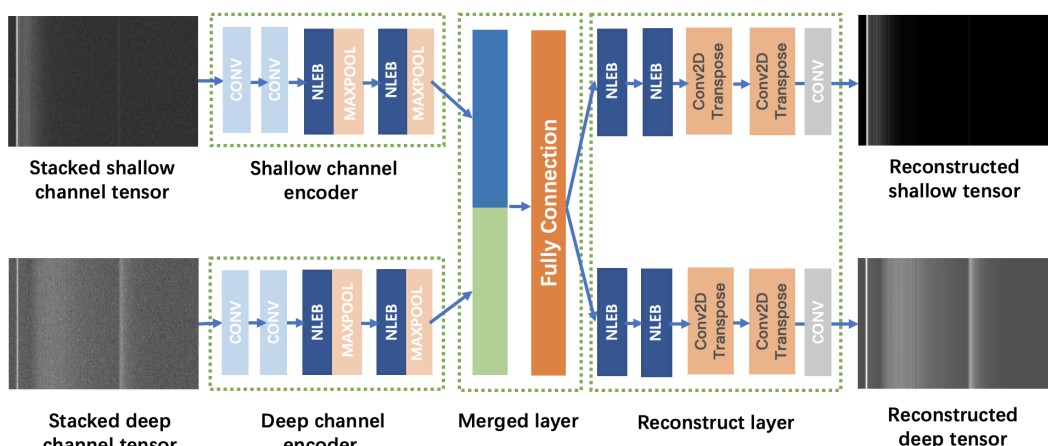

**Figure 6.** The architecture of CNLD-Net.

CNLD-Net first extracts multi-frame features through two identical branches comprising a convolution module, two NLEB modules, and two max-pooling layers. The two encoders are used to extract local and nonlocal information from deep- and shallow-water channels, respectively. The two branches' features are then flattened and concatenated as the input of two fully connected layers. We use the reconstruction layer to restore the denoised two-dimensional tensor for each channel from the shared representation. After obtaining the denoised tenors, we can obtain the denoised waveforms by detaching them. The decoding part of the CNLD network also includes two identical branches, which are composed of two NLEB modules, two deconvolution operations, and one convolutional layer. The deconvolution layer is the inverse operation of descending sampling, also known as transposed convolution, which can be used to improve the time resolution of the full waveform and restore stacked signals from the upsampling of compressed features. The dimension values after each layer in CNLD are presented in Table 2.

**Table 2.** Details for each CNLD layer.

| Module | Layer | Kernel (Size, Stride) | Dimension or Neurons $(Bs, C, L)$ [α] |
|---|---|---|---|
| Stacked image input shallow encoder /deep encoder | \ | \ | $(64,1,2000 \times 100)$ |
| | Conv-2d | $(3 \times 3,2)$ | $(64,16,1000 \times 50)$ |
| | Conv-2d | $(3 \times 3,2)$ | $(64,32,500 \times 25)$ |
| | NLEB | | $(64,64,500 \times 25)$ |
| | NLEB | | $(64,128,500 \times 25)$ |
| | Dense | | 1000 |
| Merged layer | Concatenated | | 2000 |
| Reconstruction layer | Dense | | 1,600,000 |
| | NLEB | | $(64,64,500 \times 25)$ |
| | NLEB | | $(64,32,500 \times 25)$ |
| | Conv-2d-Transpose | $(3 \times 3,2)$ | $(64,32,1000 \times 50)$ |
| | Conv-2d-Transpose | $(3 \times 3,2)$ | $(64,16,2000 \times 100)$ |
| | Conv2d | $(1 \times 1,1)$ | $(64,1,2000 \times 100)$ |
| Stacked image output | \ | \ | $(64,1,2000 \times 100 )$ |

[α] *Bs*, batch size; L, data length of the waveform; C, number of channels.

The training optimization's loss function is set as follows:

$$loss = \frac{1}{K}\sum_{i=1}^{K}(||\hat{S}_i - S_i||^2 + ||\hat{D}_i - D_i||^2) \tag{4}$$

where $S_i$ is the original data from the shallow channel, $D_i$ is the original data from the deep channel, $\hat{S}_i$ is the reconstructed tensors from the shallow channel, $\hat{D}_i$ is the reconstructed tensors from the deep channel, and K is the number of training samples of each channel. The reconstruction layer improves the time resolution by introducing a deconvolution kernel, which is also called transposed convolution. Transposed convolution fills the input features with $s - 1$ zeros between every two elements. The strides $s$ here can be understood as the magnification of the resolution, and the convolution operation with a stride of 1 is performed on the input features after filling. The width of the output features of the transposed convolution layer can be expressed as:

$$o = (w - 1) \times s - 2p + k \tag{5}$$

where $w$ represents the size of the input feature, $s$ represents the stride, $p$ represents the padding size, and $k$ represents the size of the transposed convolution kernel.

### 4.4. Performance Metrics

To evaluate the denoising ability of the proposed method, we added specified noise to measured data sets at different water depths. Then, the signals that contained noise were processed by using the proposed method, AWT [42], EMD-STRP [10], CAENN [14], 1D-Nonlocal [43], and MS-CNN [6]. Finally, the effects of the six methods were compared; the methods were evaluated by determining the following two parameters:

Root mean square error (RMSE):

$$RMSE = \sqrt{\sum_{i=1}^{N}(x_i - \tilde{x}_i)^2 / N} \tag{6}$$

Signal-to-noise ratio (SNR):

$$SNR = 10log\left(\frac{\sum_{i=1}^{N} x_i^2}{\sum_{i=1}^{N}(x_i - \tilde{x}_i)^2}\right) \tag{7}$$

Since the input and output of the network are stacked images, the parameters are calculated based on the detached waveform, where $x_i$ represents an original waveform, $\tilde{x}_i$ is an output, $N$ is the number of waveforms, and $\bar{x}$ represents the mean signal amplitude.

## 5. Experiments

### 5.1. Comparison of Signal Denoising Method

As mentioned above, the measured data were collected by the ALB system on a evening flight and at an altitude of 500 m. A total of 50,000 pieces of data with different water depths were selected, and the noise–noisier data set was constructed. We adopted the strategy of five-fold cross-validation to carry out the experiment, in which four units were selected for training and the remaining one unit was used for testing.

We chose the first 200 and the last 200 points of the signal to calculate the average power. Then, we generated random noise based on the noise coefficient and added it to the original signal to build a noise data set.

The rightmost column in the table shows the time taken to process a single signal. There is no difference in signal denoising time between the deep-water channel and shallow-water channel. The denoising time of the traditional algorithm is significantly higher than that of the deep-learning-based method. Although we extracted multi-channel and non-neighborhood features, our proposed method can still achieve high performance. With the increase in the noise coefficient, the signal-to-noise ratio decreases. Tables 3 and 4 present the RMSE and SNR of shallow-water channels and deep-water channels, respectively, with different noise coefficients. The signal-to-noise ratio of the deep-water channel after denoising is slightly higher, mainly due to the significant improvement in the seabed signal caused by high gain. The denoising algorithm based on the adaptive wavelet or soft threshold can preserve the details of the signal when processing shallow-water channel data. However, with the decrease in the signal-to-noise ratio, the problem of noise misjudgment is difficult to alleviate. Combining correlation analysis and empirical mode decomposition methods can preserve IMF components with high correlations; thus, reconstructing signals can improve the ability to retain weak signals. The signal-to-noise ratios of the CAENN and non-neighborhood architectures are significantly improved, but their RMSE values are also higher than that of the EMD-STRP algorithm due to the low stability of seabed signals in shallow-water channels and the single-frame-based network architecture. Although it is better at removing Gaussian noise, its ability to retain the target signal is not strong. The MS-CNN architecture introduces a channel attention mechanism, resulting in a significant decrease in RMSE. The proposed CNLD structure further improves the signal-to-noise ratio by considering the structural similarity of multi-frame data and the data correlation between channels.

**Table 3.** MSE and SNR values of different Gaussian noise variances in shallow channel.

| Methods | 5 | | 10 | | 15 | | 20 | | Time |
|---|---|---|---|---|---|---|---|---|---|
| | RMSE | SNR | RMSE | SNR | RMSE | SNR | RMSE | SNR | |
| AWT [10] | 0.012 | 20.55 | 0.015 | 18.62 | 0.023 | 14.95 | 0.039 | 11.51 | 0.007 s |
| EMD-STRP [42] | 0.012 | 21.03 | 0.016 | 18.55 | 0.023 | 14.96 | 0.042 | 10.59 | 0.019 s |
| CAENN [14] | 0.036 | 24.08 | 0.042 | 20.76 | 0.062 | 18.37 | 0.087 | 16.61 | 0.003 s |
| 1D-Nonlocal [43] | 0.057 | 23.94 | 0.064 | 19.84 | 0.054 | 18.42 | 0.063 | 16.91 | 0.008 s |
| MS-CNN [6] | 0.028 | 30.54 | 0.034 | 26.17 | 0.048 | 22.41 | 0.036 | 18.17 | 0.006 s |
| CNLD | 0.019 | 38.58 | 0.026 | 33.48 | 0.031 | 26.92 | 0.036 | 22.94 | 0.005 s |

**Table 4.** MSE and SNR values of different Gaussian noise variances in deep channel.

| Methods | 5 | | 10 | | 15 | | 20 | | Time |
|---|---|---|---|---|---|---|---|---|---|
| | RMSE | SNR | RMSE | SNR | RMSE | SNR | RMSE | SNR | |
| AWT [10] | 0.025 | 20.91 | 0.028 | 19.88 | 0.032 | 18.62 | 0.046 | 15.49 | 0.007 s |
| EMD-STRP [42] | 0.024 | 21.28 | 0.027 | 20.12 | 0.033 | 18.47 | 0.047 | 15.14 | 0.019 s |
| CAENN [14] | 0.052 | 32.81 | 0.054 | 30.53 | 0.057 | 28.04 | 0.064 | 22.62 | 0.003 s |
| 1D-Nonlocal [43] | 0.059 | 33.91 | 0.065 | 31.66 | 0.064 | 28.81 | 0.073 | 23.03 | 0.008 s |
| MS-CNN [6] | 0.049 | 36.82 | 0.057 | 33.48 | 0.055 | 31.62 | 0.068 | 28.18 | 0.006 s |
| CNLD | 0.027 | 40.16 | 0.045 | 37.25 | 0.052 | 33.48 | 0.062 | 29.25 | 0.005 s |

The SNR and RMSE after denoising in deep-water channels are higher than those in shallow-water channels, because a high gain simultaneously improves the signal-to-noise ratio of the target signal and noise, resulting in a decrease in denoising stability. The wavelet denoising effect of the soft threshold gradually outperforms the EMD algorithm as the noise coefficient increases, indicating that the extraction of correlation feature components in single-frame data is susceptible to Gaussian noise. The method based on deep learning is also superior to traditional algorithms in deep-water channel data denoising, with CAENN having a signal-to-noise ratio lower than 1D-nonlocal, indicating that optimizing the non-neighborhood feature extraction of time-domain signals can improve the signal-to-noise ratio of weak echoes submerged in noise. As the noise coefficient increases, the RMSE of 1D nonlocal gradually decreases compared to CAENN, indicating that non-neighborhood features exhibit high stability at low signal-to-noise ratios. The MS-CNN architecture combines multi-channel correlations to optimize the signal-to-noise ratio for denoising. Although the attention mechanism is introduced, the lack of adjacent frame features makes it unstable in preserving weak signals. Our proposed CNLD architecture not only considers the local similarity characteristics of similar positions in multiple frames of data within the channel but also considers the high-dimensional correlation of data between channels. Therefore, it can display a better denoising level with a lower root mean square error and the lowest signal-to-noise ratio. Below, we further compare the denoising effects by visualizing stacked echoes and single-frame echoes, with the shallow-water channel results at the top and deep-water channel results at the bottom of each grouping.

Figure 7a,b show the stacked noise echoes and pure echoes, respectively. The two white vertical lines, representing the sea-surface and sea-bottom signals, indicate that the preprocessed stacked echo has relevance. The darker the color in the stacked image, the lower the noise, and it is obvious that the superimposed noise echoes are generally gray. Compared to Figure 7b,c, it can be seen that the color of EMD-STRP after denoising is darker, indicating a more significant effect on Gaussian noise. By comparison with Figure 7e,f, it can be seen that the convolutional neural network based on the encoder–decoder can remove Gaussian noise by relying on multi-layer convolution to extract the high-dimensional features of echoes, but the white part of CAENN presents obvious distortion. However, 1D nonlocal networks retain more noise while preserving the target signal without deformation. MS-CNN in Figure 7g significantly improves the ability to remove Gaussian noise and retains effective signals by introducing attention modules for deep- and shallow-water channels. However, due to the high randomness of laser sea survey data, the channel correlation characteristics will also lead to denoising distortion. As shown in Figure 7h, the denoising result of the proposed method is most similar to the ground truth. By extending the receptive field in the spatial direction, the correlation of stacked signals of the deep-water channel and shallow-water channel can be fully exploited. Therefore, our proposed method can distinguish low-reflectivity targets from noise, reduce noise near the right white line, and preserve the seafloor signal.

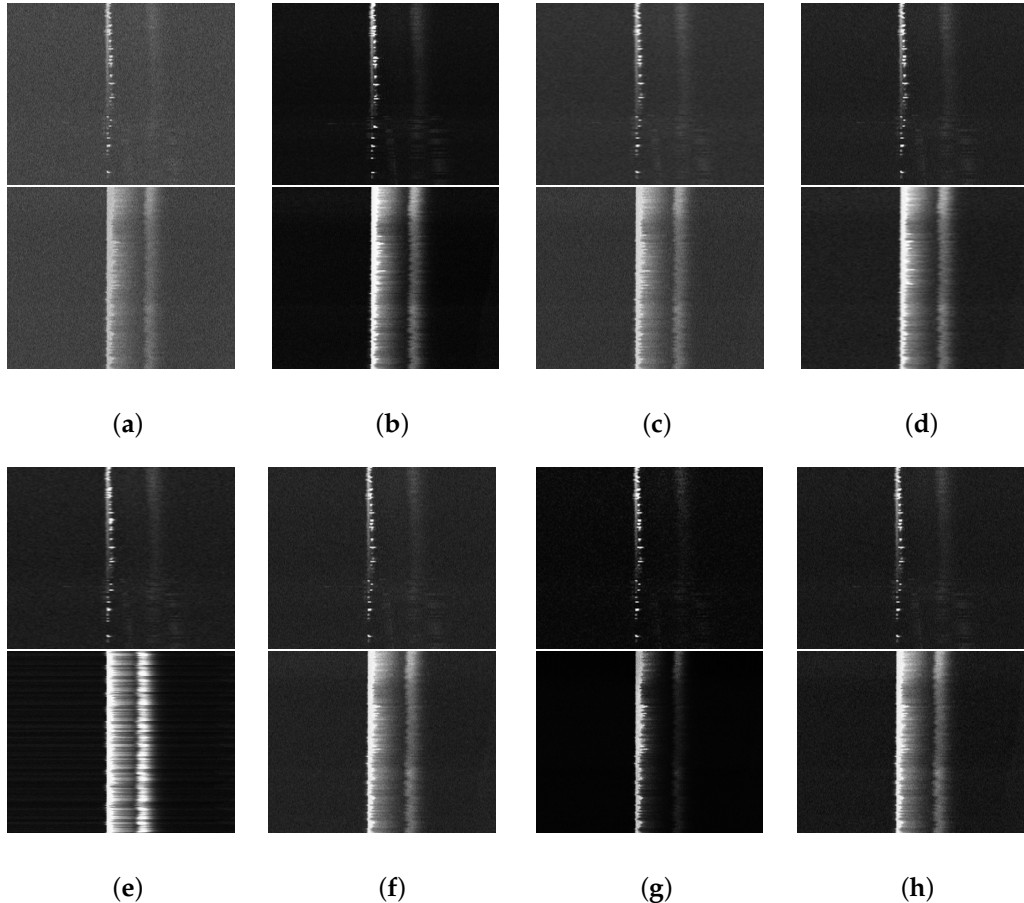

**Figure 7.** Visual comparison of measured stacked images. (**a**) Noise signal; (**b**) ground truth; (**c**) AWT; (**d**) EMD-STRP; (**e**) CAENN; (**f**) 1D-Nonlocal; (**g**) Ms-CNN; (**h**) CNLD.

We selected a frame of signals with obvious sea-surface and sea-bottom echoes in deep- and shallow-water channels for visual comparison to verify the ability of the proposed algorithm to retain the sea-bottom signal. As shown in Figure 8, the interference noise of LiDAR echo signals mainly includes system noise generated by internal hardware, as well as non-system noise generated by external environmental factors, such as the atmosphere and water. We selected one frame of echoes for visualization; the backscatter between the sea-surface and seabed signals was received by deep-water channels with a larger field of view. Figure 8a,b are, respectively, the original signal after adding a noise factor of 25 and the pure signal. Figure 8c,d demonstrate the denoising result of dynamic frequency-domain filtering and EMD-STRP. Although Gaussian noise can be removed by local refinement, the weak signal on the seabed is still confused with noise. As shown in Figure 8e,f, data-driven denoising methods can learn high-dimensional temporal features in echoes and thus can better remove Gaussian noise. However, the scattering of seawater and the low reflectivity of the seafloor will lead to the distortion of the weak seafloor signal during denoising. By introducing one-dimensional nonlocal features, the aliasing of Gaussian noise and of the target signal can be further alleviated. Figure 8g has a lower noise signal-to-noise ratio, mainly reflected in the smoother bottom noise of the signal and the better fit to the ground truth after denoising near the peak of the dual-channel seabed signals. It can improve the ability to retain weak echoes but also retains more jitter. As shown in Figure 8h, our proposed method makes use of the multi-frame correlation to improve the signal-to-noise ratio of seafloor signals. By taking into account the characteristics of multi-channel signals, CNLD retains more details and improves the stability of denoising. In general, the proposed method has a higher signal-to-noise ratio and a lower MSE than other methods, achieving the best denoising effect.

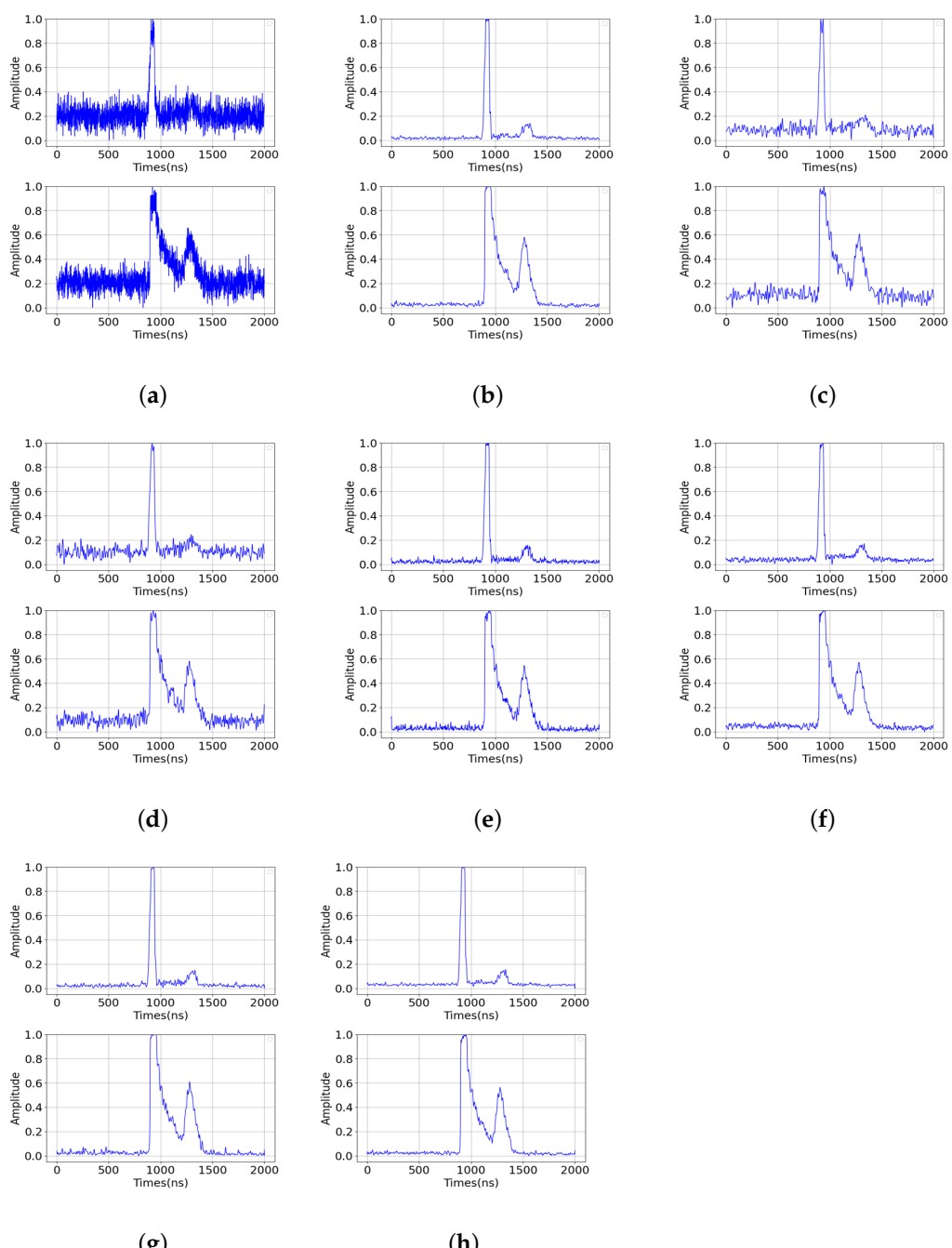

**Figure 8.** Visual comparison of measured waveforms. (**a**) Noise signal; (**b**) ground truth; (**c**) AWT; (**d**) EMD-STRP; (**e**) CAENN; (**f**) 1D-NONLOCAL; (**g**) Ms-CNN; (**h**) CNLD.

## 5.2. Comparison of Different Dilation Rates

The deep-water channel and shallow-water channel of multi-channel airborne LiDAR share the same optical system, but the field-of-view angle and detector gain are different. Therefore, there is a correlation between echoes from different channels and between adjacent frames. We designed various parameters to compare the effects of perceptual spatial fields. When multiple layers of dilated convolution are introduced, local information may be lost due to the gridding effect. Therefore, an HDC (hybrid dilated convolution) architecture [44] is referred to in the design of dilation rates for multi-layer dilated convolution. The expansion rate of stack unwrapping convolution cannot have a common divisor greater than 1, so the characteristics of local and nonlocal echoes can be comprehensively consid-

ered. We adopted three-layer stacked convolution as the basic architecture of the NLEB module. As shown in Figure 9a, the receptive fields of three conventional convolution stacks with a dilation rate of 1 can be regarded as traditional convolution stacks. As shown in Figure 9b,c, dilation rates of 1,2,3 and 1,2,5, respectively, in the spatial direction can expand the receptive field. In order to explore the ability of the proposed network to extract spatial features, we used dilated convolution in both spatiotemporal directions and only in the spatial direction.

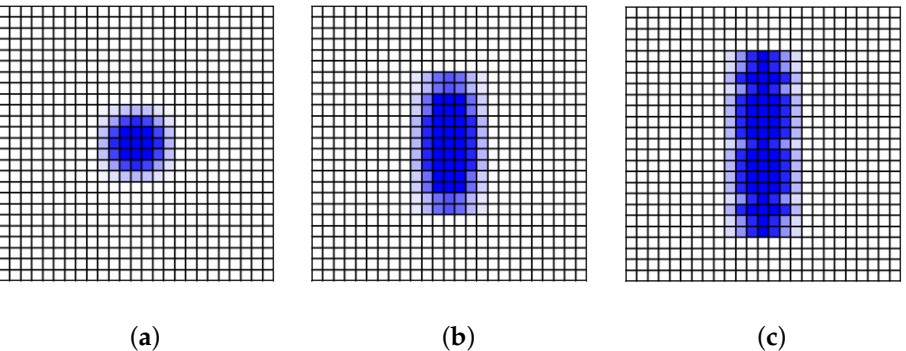

(**a**) (**b**) (**c**)

**Figure 9.** Comparison of different dilation rates in spatial direction. (**a**) Dilation rates = (1,1,1); (**b**) dilation rates = (1,2,3); (**c**) dilation rates = (1,2,5).

As shown in Table 5, with the increase in the dilation rate in the convolution layer, the signal-to-noise ratio of (1,2,2) is significantly higher than that of (1,1,1). Using dilated convolution can enlarge the local receptive field, so the signal-to-noise ratio is more reliable. When dilated convolution is used in both directions, the signal-to-noise ratio does not improve significantly and even becomes lower. This is because introducing dilated convolution in the time dimension can increase the receptive field, but it will destroy the dependency of the time series. More adjacent echoes can be covered by setting the dilation rates of the three stacked convolution layers to 1, 2, and 5. However, the improvement effect is not obvious when compared with setting the dilation rates to 1, 2, and 3. This is because the correlation or similarity of full-waveform data also decreases with the increase in the spatial distance. Using a large-scale convolution kernel to extract time-series features can improve the extraction ability of local features, but it will reduce the efficiency of network convergence. Therefore, the NLEB module proposed in this paper can enhance the efficiency of extracting the similarity of multi-frame echoes by using stacked small-kernel dilated convolution.

**Table 5.** Denoising effect of different dilation rates in measured experiment.

| SNR | Evaluation | Non | Dilation Rates in Spatial Direction | | | Dilation Rates in Both Directions | | |
|---|---|---|---|---|---|---|---|---|
| | | (1,1,1) | (1,2,2) | (1,2,3) | (1,2,5) | (1,2,2) | (1,2,3) | (1,2,5) |
| 5 | RMSE | 0.041 | 0.035 | 0.023 | 0.022 | 0.033 | 0.029 | 0.031 |
| | SNR | 37.22 | 38.68 | 39.37 | 39.32 | 38.74 | 39.28 | 39.33 |
| 10 | RMSE | 0.047 | 0.042 | 0.036 | 0.035 | 0.043 | 0.047 | 0.049 |
| | SNR | 33.66 | 34.51 | 35.36 | 35.33 | 34.92 | 35.27 | 35.32 |
| 15 | RMSE | 0.038 | 0.040 | 0.041 | 0.043 | 0.041 | 0.046 | 0.044 |
| | SNR | 28.81 | 29.76 | 30.18 | 29.86 | 29.95 | 30.19 | 30.20 |
| 20 | RMSE | 0.043 | 0.045 | 0.048 | 0.050 | 0.046 | 0.051 | 0.049 |
| | SNR | 25.47 | 25.63 | 26.09 | 25.87 | 25.94 | 26.01 | 26.02 |

### 5.3. Ablation Experiment

To verify the contribution of the multi-channel correlation and the proposed NLEB module to noise reduction, we designed Baseline-Net. As shown in Figure 10, this architecture uses our proposed NLEB module to extract features and uses a deconvolution module to recover signals.

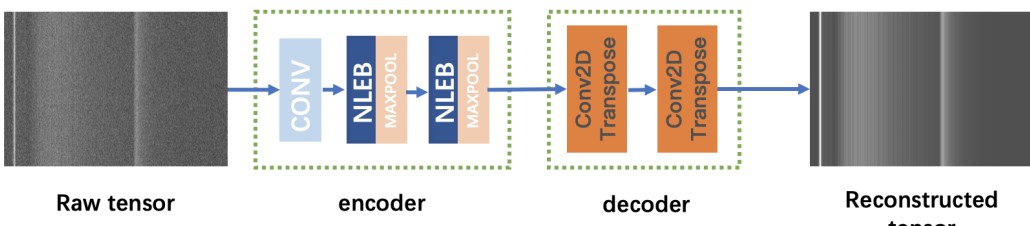

**Figure 10.** The architecture of Baseline-Net.

As shown in Table 6, under the condition of the same noise coefficient, the multi-frame deep-water echo after stacking has a stronger detector gain and a higher signal-to-noise ratio, but the RMSE of the shallow-water channel is lower and the denoising stability is better. Due to the adjustment of the gain and field-of-view angles in deep- and shallow-water channels, there are some blind spots in seabed exploration. Therefore, the signal-to-noise ratio of CNLD-Local is significantly higher than the baseline single-channel denoising effect. After the introduction of the proposed NLEB module, the RMSE of the CNLD model is significantly lower than that of the CNLD-Local model, indicating that the enhancement of the receptive field makes the denoising model more robust.

**Table 6.** Ablation experiment using measured data.

| Signal | SNR_Noise | Baseline | | Baseline | | CNLD-Local | | CNLD | |
|---|---|---|---|---|---|---|---|---|---|
| | | RMSE | SNR | RMSE | SNR | RMSE | SNR | RMSE | SNR |
| Shallow channel | 5 | 0.038 | 34.91 | - | - | 0.031 | 36.25 | 0.021 | 38.58 |
| | 10 | 0.029 | 28.29 | - | - | 0.041 | 31.63 | 0.026 | 33.48 |
| | 15 | 0.051 | 22.71 | - | - | 0.038 | 24.65 | 0.031 | 26.92 |
| | 20 | 0.069 | 20.29 | - | - | 0.043 | 22.27 | 0.036 | 22.94 |
| Deep channel | 5 | - | - | 0.053 | 37.65 | 0.049 | 38.19 | 0.027 | 40.16 |
| | 10 | - | - | 0.075 | 34.69 | 0.054 | 35.69 | 0.045 | 37.25 |
| | 15 | - | - | 0.082 | 30.32 | 0.057 | 32.95 | 0.052 | 33.48 |
| | 20 | - | - | 0.088 | 28.37 | 0.072 | 28.56 | 0.062 | 29.25 |

### 5.4. Analysis of Limitations

Because of rapid changes in the environment, not all data between scene channels have a strong correlation. By further analyzing the limitations of the laser sea denoising algorithm through stacked echo diagrams, it can be seen that sea-surface and seabed signals have strong randomness. As shown in Figure 11a, the positions of the sea surface and seafloor change between multiple frames, but their relative positions remain unchanged, showing a good correlation. As shown in Figure 11b, due to the fast flight of the aircraft or sea-surface jitter, the fast transformation of the target signal will cause the correlation to weaken after multi-frame stacking. Although the method proposed in this paper uses the centroid of the sea surface to detect the signal, no alignment method can optimize the denoising effect by using the echo similarity after the sea surface is lost. As shown in Figure 11c, the sea-surface signal has strong randomness, and the field-of-view angle of the deep-water channel is larger, resulting in a higher probability of receiving the target signal. However, shallow-water channels are limited by the field-of-view angle scale and gain, which reduces the probability of receiving effective signals. As shown in Figure 11d, when

the laser hits trees, multiple targets will be generated. However, the gain and sensitivity of deep- and shallow-water channels to targets are different, so when the echoes appear saturated or there are multiple targets, the correlation of data between channels will also decrease.

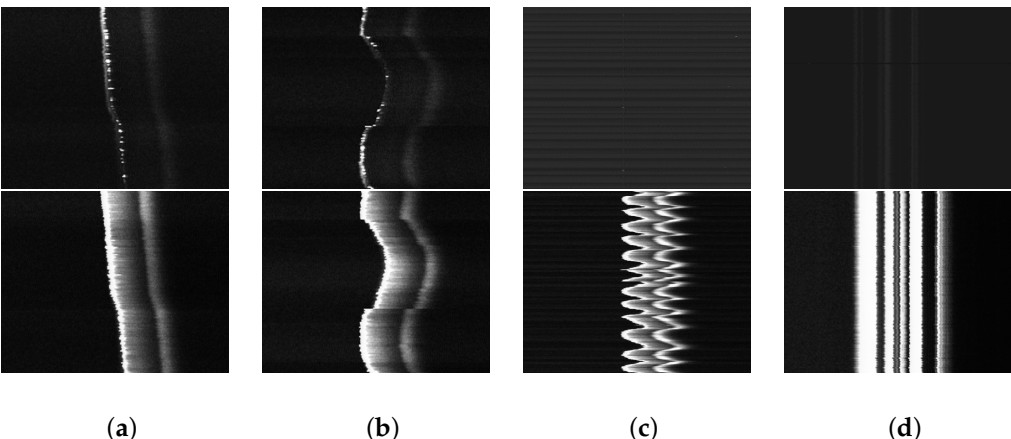

(**a**)         (**b**)         (**c**)         (**d**)

**Figure 11.** Visual comparison of stacked images. (**a**) Strong correlation; (**b**) weak correlation; (**c**) channel anomaly; (**d**) multiple echoes.

As shown in Figure 12, limiting denoising factors are analyzed through the visualization of a single waveform. As shown in Figure 12a, the correlation of echoes is mainly reflected in sea-surface and seabed signals. After normalization, seabed signals are significantly different due to the different gains of deep- and shallow-water channels. The field of view of the deep-water channel is large, so more water body backscattering can be received, resulting in the deformation of seabed signals. As shown in Figure 12b, hardware circuit overheating, random background light, and other factors may lead to the availability of data in a single channel in some cases, so the data correlation between channels may fail in some cases. As shown in Figure 12c, due to the difference in gains of deep- and shallow-water channels, the target signal of the shallow-water channel is normal when the deep-water channel is saturated, and the deep-water channel will receive more target signals due to a stronger gain.

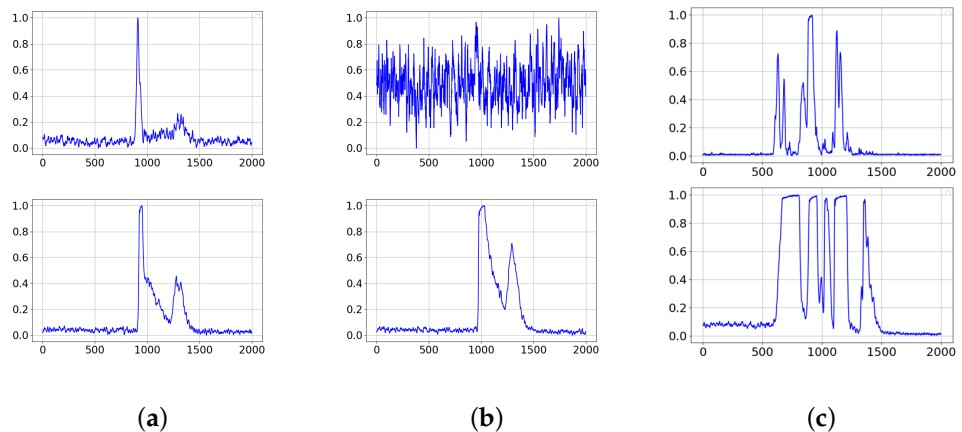

(**a**)         (**b**)         (**c**)

**Figure 12.** Visual comparison of single waveform. (**a**) Weak correlation; (**b**) channel anomaly; (**c**) multiple echoes.

## 6. Conclusions

In this paper, a denoising method for a multi-channel ALB full waveform is proposed. Firstly, an alignment method based on surface centroids was used to highlight the structural similarity between consecutive frames. Then, we introduced the NLEB module to enhance the spatial receptive field, and the correlation extraction efficiency of the sea-surface and seabed echoes was optimized. Finally, we designed a CNLD network based on a multitasking learning architecture to mine signal correlations between channels and simultaneously reconstruct denoised deep and shallow channel echoes in an end-to-end manner.

Through ablation experiments, we found that the correlation between multiple waveforms in the same channel exhibits high stability, and the correlation between channels is constrained by channel gain differences, field-of-view angle differences, and the signal's dynamic range. However, the signal correlation between channels can still improve the retention of weak seafloor signals. The proposed method can obtain the highest SNR with the lowest RMSE under different noise variances. In the future, we will further optimize the method of extracting channel correlations and improve the robustness of the model through data enhancement and the introduction of prior physical knowledge.

**Author Contributions:** Conceptualization, Y.Z.; Methodology, B.H.; Validation, J.H. and Q.L.; Investigation, C.L.; Resources, M.Y. and Y.L.; Funding acquisition, G.Z. All authors have read and agreed to the published version of the manuscript.

**Funding:** This work is supported by the Guangxi Innovative Development Grand Grant (No. 2018AA13005).

**Data Availability Statement:** The data presented in this study are available on request from the corresponding author. The data are not publicly available due to project confidentiality agreement.

**Conflicts of Interest:** The authors declare no conflict of interest.

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
