# Peer review of "Coupling Dilated Encoder–Decoder Network for Multi-Channel Airborne LiDAR Bathymetry Full-Waveform Denoising"

_remotesensing, doi:10.3390/rs15133293_

Round 1

Reviewer 1 Report

This article is about airborne bathymetric full waveform denoising. The signal-to-noise ratio of airborne full waveform is extremely unstable, especially when it comes to underwater detection, the complex transmission process will further increase the uncertainty of weak seabed echo signals. Traditional multi-frame stacking may lead to weak signals being overly smoothed and missed. Overall, the author's approach of viewing multi-frame signals as images is very innovative. Combining multi-task learning architecture and dilated convolution, the denoising efficiency and signal-to-noise ratio of multi-channel airborne lidar bathymetric echoes are improved, making full use of the correlation of multi-channel echoes.

1.        The references are sufficiently cited. It is suggested to strengthen the analysis of the advantages and disadvantages of different methods in the related work, and the language expression needs to be optimized.

2.        The analysis of the characteristics of ALB data is very clear. The characteristics of different channel echoes are analyzed through formulas and waveform diagrams, but the data selected in Figure 8 need to be refined. What single-frame echo was used under what circumstances.

3.        It is recommended to add the implementation method and formula description of deconvolution in the method description and refine the description of signal recovery.

4.        The purpose of this paper is to improve efficiency. Is there any optimization after introducing multi-channel and multi-frame processing? It is recommended to add an experiment on denoising time in the comparison.

5.        The calculation of the noise coefficient in the comparative experiment needs to be more detailed, such as pointing out the 5,10,15,20 in Tables 3 and 4.

6.        The language in the conclusion section is quite general and not specific enough, it is recommended to introduce the description of indicators.

Author Response

Dear Editors and Reviewers, The authors would like to thank the editors for coordinating the review of this manuscript and the kindness in offering the chance to revise. The authors have made the best effort to address all the issues raised by the reviewers and revised the manuscript thoroughly and carefully. The authors would like to thank reviewers for their constructive comments, which have helped the authors improve the quality of the revised manuscript. The point-by-point responses to the comments of each reviewer are provided below, and all the major changes are highlighted in blue color in the revised manuscript.

Reviewer 2 Report

Coupling Dilate Encoder-Decoder Network for Multi-channel Airborne LiDAR Bathmetry Full-waveform Denoising

In this manuscript, the authors propose a non-local encoder block (NLEB) based on spatial dilated convolution to optimize the feature extraction of adjacent frames in full waveform LiDAR. For that, they propose a coupled denoising encoder-decoder network, which takes advantage of echo correlation in deep water and shallow water channels. The results show that the proposed method improves the stability of denoising by using inter-channel and multi-frame data correlation.

This manuscript is interesting and I have enjoyed and learned at the time I was reading that. However, I have some concerns. I can recommend this manuscript for publication once the following concerns are appropriately addressed by the authors:

·         The authors argue “Firstly, the echoes from different channels are separately stacked together to form two-dimensional tensors that are fed into the proposed network” -> (a) replace the expression “that are fed into” for another one more appropriate for scientific writing; (b) What the authors mean with “echoes” if we are talking about full-waveform? Do these systems emit individual/discrete pulse beams? Do you refer to the multiple impacts/hits from a same unique laser beam?

·         In the abstract, also the authors argue: “Weizhou Island measured waveform were used for denoising comparison and ablation experiments” -> I do not understand this sentence.

·         Along the manuscript the authors argue “Airborne full-waveform lidar is” ->  (a) The authors should be more consistant with the different names and concepts. It is commonly used the acronyms of ALS (Airborne Laser System) or ALTM (Airborne Laser Terrain Mapping); (b) The same happen with “lidar” -> The authors should use “LiDAR” instead.

·         When the authors introduce the system they argue “which can store and record the complete reflected echo signal of the target in a very short sampling interval and a large capacity, so it can obtain a high range resolution [1]”. The study number 1 refers to a relatively recent study (Ke and Lam, 2018). My suggestion to the authors is to introduce clearly in two paragraphs the ALS/ALTM/LiDAR technology showing the most relevant facts related to that. Of course, I can recommend you to include the works from Baltsavias (1999) “Airborne laser scanning: basic relations and formulas”, but also other ones referring to the importance of different scanning patterns (Balsa, Avariento and Lerma, 2012: “Airborne light detection and ranging (LiDAR) point density analysis”), and other relevant factors. Just after that, the authors should start with the particular case of the full-waveform LiDAR.

·         In the list of keywords the authors refer to “gramian angular gield” -> Is this concept well expressed?

·         In the introduction, the authors elaborate a list of potential applications: “Airborne Lidar is widely used in land cover classification, Marine resource detection, vegetation detection, military target detection and other fields because of its high measuring efficiency, wide working range, high measuring accuracy and the ability to obtain the physical characteristics of the target”. -> (a) Why “Marine” is on initial’ capital letter? (b) The list is quite incomplete and it is not exhaustive; (c) The authors should include some potential studies with regard to each application. For example, with regards to land cover classification I suggest you the study from Hermosilla et al (2012; “Land-use Mapping of Valencia City Area from Aerial Images and LiDAR Data” / 2014: Using street based metrics to characterize urban typologies”)

·         After that, the authors argue “Because 532nm blue green light has good water permeability, the laser radar of this wavelength is used to measure the depth of the sea floor”. -> All the bathymetric LiDAR use the same wavelength?

·         Since the laser beam from the atmosphere into the sea surface and travels through the water, the energy returned to the detector will be reduced, which will reduce the peak signal-to-noise ratio of the seafloor signal and increase the uncertainty of distance settlement[2]” -> Can the authors talk about accuracy/precision/integrity of this signal?

·         Gaussian noise and abrupt noise will be introduced by sea surface fluctuation, aircraft jitter, water backscattering, and other factors, so the design of special signal denoising technology for the airborne lidar bathymetry(ALB) system is still a research hotspot”. I recommend to rephrase this sentence because it is just pretty unclear.

·         In Figure 2, (a) What are the meaning and units used in both axis?; (b) What does it mean each coloured-lane represented?

·         In Figure 3, the authors must present a spatial scale and legend.

·         The authors should review exhaustively the manuscript because I have observed some typos and inaccuracies in the text. For example, “Bathmetry”, “lidar”, or the inconsistent use of some expressions and words.

Author Response

(The authors gave the same response as above.)
